# Tunneling Spectroscopy for Electronic Bands in Multi-Walled Carbon Nanotubes with Van Der Waals Gap

**DOI:** 10.3390/molecules26082128

**Published:** 2021-04-07

**Authors:** Dong-Hwan Choi, Seung Mi Lee, Du-Won Jeong, Jeong-O Lee, Dong Han Ha, Myung-Ho Bae, Ju-Jin Kim

**Affiliations:** 1Department of Physics, Jeonbuk National University, Jeonju 54896, Korea; cdh1183@skku.edu (D.-H.C.); duwon@krict.re.kr (D.-W.J.); 2Korea Research Institute of Standards and Science, Daejeon 34113, Korea; seungmi.lee@kriss.re.kr (S.M.L.); dhha@kriss.re.kr (D.H.H.); 3Korea Research Institute of Chemical Technology, Daejeon 34114, Korea; jolee@krict.re.kr; 4Department of Nano Science, University of Science and Technology, Daejeon 34113, Korea

**Keywords:** van-der-Waals gap, tunneling spectroscopy, multi-walled carbon nanotubes, indium

## Abstract

Various intriguing quantum transport measurements for carbon nanotubes (CNTs) based on their unique electronic band structures have been performed adopting a field-effect transistor (FET), where the contact resistance represents the interaction between the one-dimensional and three-dimensional systems. Recently, van der Waals (vdW) gap tunneling spectroscopy for single-walled CNTs with indium–metal contacts was performed adopting an FET device, providing the direct assignment of the subband location in terms of the current–voltage characteristic. Here, we extend the vdW gap tunneling spectroscopy to multi-walled CNTs, which provides transport spectroscopy in a tunneling regime of ~1 eV, directly reflecting the electronic density of states. This new quantum transport regime may allow the development of novel quantum devices by selective electron (or hole) injection to specific subbands.

## 1. Introduction

The van der Waals (vdW) contacts between metals and low-dimensional materials do not disturb the physical properties of the low-dimensional materials at the interface [1,2,3]. For the vdW metal–semiconducting transition metal chalcogenide (TMDC) junctions, the non-perturbed interface allows a tunable Schottky barrier height with various metals having different work functions, following the Schottky–Mott rule [1]. In this case, to achieve vdW contacts between metals and TMDC, metals are mechanically transferred to the TMDC flakes. In particular, it has been revealed that thermally evaporated In metal on TMDCs can also provide the vdW interface with good contact resistance, contrary to other metals evaporated with high evaporation energy, inducing atomic defects in the TMDC layers [2,3,4,5]. For single-walled carbon nanotubes (SWCNTs), the In/SWCNT vdW interface functions as a vacuum gap, allowing vdW gap tunneling spectroscopy for the electronic band in SWCNTs [6].

In this work, we extend the vdW gap tunneling spectroscopy to multi-walled CNTs (MWCNTs): two semiconducting MWCNTs and one metallic MWCNT. Low-temperature transport measurements are performed with indium (In)-contacted MWCNTs to reveal multiple conductance peaks and dips that are interpreted as the electronic density of states (DOS) of MWCNTs. This implies that vdW gap tunneling spectroscopy with In electrodes for CNTs provides a feasible way to assign the subbands during operation of a CNT field-effect transistor (FET).

## 2. Results and Discussion

### 2.1. vdW Gap Tunneling Spectroscopy

Figure 1a presents schematic illustrations of the electronic DOS as a function of the electronic energy in a semiconductor (*sm*)-MWCNT (the middle panel) with source and drain electrodes. Here, SC*n* (SV*n*) is the *n*th subband in the conduction (valence) band in the *sm*-MWCNT, which corresponds to the van Hove singularity in the DOS of CNT. The Fermi level of the MWCNT (*E*_FC_) is located near the conduction band edge (i.e., SC1) in the bandgap. The Fermi levels of the source (*E*_FS_) and drain (*E*_FD_) electrodes are also aligned with *E*_FC_ under zero bias conditions (the dashed vertical line in the source and drain electrode diagram). The main difference from a conventional field-effect transistor (FET) with Ohmic contacts is the existence of the vdW gap between the source (drain) and MWCNT. In this case, a negative source–drain bias voltage (*V*_sd_) applied to the source electrode shifts *E*_FS_ to a higher energy, causing *E*_FS_ to become aligned with SC1, as indicated by the red line in the source electrode diagram. Then, electrons in the source electrode start to tunnel into the MWCNT though the vdW gap, resulting in a current (*I*) flow, and the differential conductance (d*I*/d*V*_sd_) at a given *V*_sd_ is proportional to the DOS in the MWCNT. By scanning *V*_sd_, one can obtain the DOS (*E*) for the valence and conduction bands.

### 2.2. Characterization of DWCNT

To realize the vdW gap between the metal and MWCNT, we deposit 60 nm thick In metal for the source and drain electrodes on a double-walled CNT (DWCNT) located on a 500 nm thick SiO_2_/Si substrate with a back-gate geometry, as shown in Figure 1b. During the deposition, the substrate stage in the vacuum chamber was kept at 100 K with liquid nitrogen to obtain homogeneity of the In film [6]. The DWCNT channel length (*L*) was ~6 μm. The homogeneity of the DWCNT was checked by the Raman spectra at two points ~2 μm apart. For instance, Figure 1c shows the Raman spectra of (i) and (ii) obtained from the dashed circled regions indicated by (i) and (ii) in Figure 1b, respectively, with a 514 nm wavelength laser. Both spectra show a main G peak at 1598 cm^−1^ with a satellite peak at 1586 cm^−1^ (see two vertical dashed lines). This is consistent with a previous report presenting DWCNTs with a similar laser wavelength [7]. Figure 1d shows the height profile obtained from the AFM scanning along the direction of the arrow in Figure 1b, which provides the diameter (*D*) of the DWCNT, ~1.7 nm.

### 2.3. vdW Tunneling Spectroscopy for DWCNT

Figure 2a shows the transfer curve, current (*I*) as a function of back-gate voltage (*V*_bg_) at source-drain voltage (*V*_sd_) of 2 V for the DWCNT, which exhibits semiconducting (*sm*) behavior at *T* = 10 K. The d*I*/d*V*_sd_ map for the *sm*-DWCNT in Figure 2b shows that the DWCNT is slightly intrinsically hole(*h*)-doped. *V*_sd_, corresponding to the multiple conductance peaks observed for *V*_bg_ < −20 V in Figure 2b, seems to be insensitive to *V*_bg_. However, near *V*_bg_ = −15 V, the lines corresponding to the peaks abruptly curve to lie parallel to the depletion boundary that is indicated by the left-hand white dashed line in the yellow oval. In Figure 2c, when *V*_bg_ decreases across *V*_bg_ = −20 V, an additional conductance peak appears at *V*_sd_ = 0.32 V, as indicated by the dotted circle at *V*_bg_ = −25 V. This corresponds to the newly appeared conductance peak curve for *V*_bg_ < −20 V, as indicated by an arrow in Figure 2b. This implies that a subband with a relatively lower index number can be detected at a sufficiently high negative *V*_bg_ region. This is because a relatively large bandgap causes noticeable charge depletion in the CNT near the metal interface with the corresponding bias conditions (*V*_bg_ ~ −20 V), which suppresses the tunnel probability. To understand this, we construct a band model, as described in the following section.

### 2.4. Band Model for sm-DWCNT Tunnel Junction

The slightly *h*-doped state indicates that the *E*_F_ of the source (S) and drain (D) electrodes are located at a lower location from the mid-gap (*E*_g_/2) of the *sm*-CNT (see the vertical line in Figure 2b). In the gap region, Figure 2b shows zero conductance for examined *V*_sd_ up to ±2 V at −10 V < *V*_bg_ < 20 V. Near *V*_bg_ ~ 7 V, the bands are nearly flat (see the left panel of Figure 3a). With positively increasing *V*_bg_, the bands become pulled towards a lower direction while the ends of the CNT bands keep their original positions as shown by the green curves in the middle panel of Figure 3a [8], where only the lowest (SC1) and highest (SV1) bands of the conduction and valance bands are depicted, respectively. In this case, although a positive *V*_sd_ comparable to *E*_g_ is applied, currents cannot flow through the CNT due to the thick potential barrier indicated by the dashed green circle for the green band in the right-hand panel of Figure 3a. For *V*_bg_ > 20 V in Figure 2b, the bands in the CNT are significantly bent, as shown by the red curves in the middle of Figure 3a. At the same *V*_sd_ condition as the green bands, electrons then encounter a relatively thin potential barrier, as indicated by the red dashed circle for the red bands in the right-hand panel of Figure 3a. This allows the current to flow through the DWCNT and forms the conductance boundary indicated by the dashed yellow curve in Figure 2b.

In a hole-doped region for *V*_bg_ < 7 V, the DWCNT bands are pulled towards an upper direction. For *V*_bg_ ~ −10 V in Figure 2b, the maximum energy of the valence band is still lower than the *E*_F_ of the electrodes, as shown in the left-hand panel of Figure 3b. With positively increasing *V*_sd_ in the middle panel of Figure 3b, the potential barrier width at the source becomes narrower and a tunnel event occurs. The middle of Figure 3b shows a case of detectable tunneling current when *E*_FS_ is entered into a certain subband with a sufficiently positive *V*_sd_ (see the red arrow in the middle panel of Figure 3b), which will produce a conductance peak, as observed at *V*_sd_ ~ 0.5 V for *V*_bg_ = −17 and −20 V in Figure 2c. We found an irregular conductance region in the dashed box at *V*_bg_ = −20 V in Figure 2c, located at a lower *V*_sd_ than that of the main peak at *V*_sd_ ~ 0.5 V. This could be related to a lower subband. For instance, the tunneling probability through the low-index subband with a relatively low *V*_sd_ value would be too low to measure a pronounced conductance peak, due to the relatively thick potential barrier. If a negative *V*_sd_ is applied near *V*_bg_ ~ −20 V, the Zener tunneling current (green arrow) with the general tunneling current (blue arrow) becomes involved, as shown in the right-hand panel of Figure 3b, resulting in the smearing of local conductance peaks for negative *V*_sd_.

With more negative *V*_bg_ in Figure 2b, the bands are more bent and the maximum of the valance band more approaches the Fermi levels of the electrodes, as depicted in the left-hand panel of Figure 3c. In this case, it could provide a detectable tunneling current when *E*_FS_ is aligned to the low subband with a positive *V*_sd_. This change is indeed shown in Figure 2b, indicated by the vertical arrow at *V*_bg_ = −22 V, where a new conductance peak curve is introduced with negatively increasing *V*_bg_. At *V*_sd_ = −25 V in Figure 2c, the new peak appears in the irregular conductance region, thus we consider that the new peak should have a lower subband index number than that for the peak at *V*_bg_ = −20 V. For a negative *V*_sd_, the tunneling current through the subbands in the conduction band is also expected before entering into the Zener regime because the tunnel barrier is already narrowed, as shown in the right-hand panel of Figure 3c. With negatively increasing *V*_bg_, the new peak follows the conductance curve indicated by the arrow in Figure 2b and corresponds to the peak indicated by the red arrow at *V*_sd_ ~ 0.25 V and *V*_bg_ = −40 V in Figure 2c. Since the bands become bent towards the upper direction with negatively increasing *V*_bg_, the tunnel probability becomes higher for negatively increasing *V*_bg_. Then, there should be finite conductance for *V*_sd_ < 0.25 V if there are low-index subbands, as in the irregular conductance region in the dashed box at *V*_bg_ = −20 V. However, we could not identify a noticeable finite conductance in the *V*_sd_ region, which suggests that the peak indicated by the red arrow corresponds to the lowest subband, SV1. Then, the peak at *V*_sd_ ~ 0.5 V for *V*_bg_ = −20 V can be considered as SV2. Following a similar analysis, it is suggested that the peak indicated by the blue arrow at *V*_bg_ = −40 V corresponds to SC1.

### 2.5. Density Functional Theory (DFT) Calculations for DWCNTs

We consider that the tunneling between the electrode and DWCNT occurs based on the hybridized electronic band structure between inner and outer CNTs [9,10,11]. In order to theoretically determine the electronic structure of DWCNTs, we performed DFT calculations with five different SWCNTs, the components of DWCNTs, as listed in Table 1. Although the DFT method has been known to underestimate the electronic gap when using a “conventional” exchange-correlation functional of local density approximation (LDA) or GGA functional, the comparative study of the *E*_g_ of similar structures, such as CNTs, is still considered practical and relevant. Table 1 shows the chiralities, diameters, the calculated *E*_g_ and the electrical properties of SWCNTs. We selected the potential outer tube with a diameter of 1.3 ~ 1.5 nm and then selected the relevant inner tube. Only zigzag tubes were considered due to their various electronic properties according to the chirality and also the relatively short repeating unit length in calculational geometry along the tube axis. The calculated *E*_g_ values show the expected well-known trends, which are semiconducting for (3*n*+1, 0) or (3*n*+2, 0) and metallic for (3*n*, 0) SWCNTs. The small bandgap for (9, 0) originated from the large π*-σ* hybridization in CNTs with a small diameter, as reported previously [12]. This π*-σ* hybridization effect becomes negligible as the diameter increases, as we can see clearly when comparing the *E*_g_ of (9, 0) and (18, 0) SWCNTs listed in Table 1. In order to interpret our experimental observations, we calculated the electronic properties of DWCNTs. Using the SWCNTs listed in Table 1, we generated four DWCNTs with properties of *sm*@ *sm*, *m*@ *sm*, *sm*@*m* and *m*@*m* (inner@outer CNTs), keeping the intertube distance close to the van-der-Waals distance, where *m* indicates a metal. The calculated electronic bandgap and the properties are listed in Figure 4 and Table 2. For the *sm*@*sm* case, *E*_g_ did not change significantly from the original value. However, in the case of *m*@*sm*, *E*_g_ was suppressed by as much as a factor of two, although it retained the semiconducting property, which was due to the overlapping of the electrons’ wavefunctions in the metallic and semiconducting CNTs. On the other hand, both *sm*@*m* and *m*@*m* showed metallic behaviors while retaining the electronic structure of the inner CNT. A recent systematic study showed that the *E*_g_ of DWCNTs is smaller than the average value of the *E*_g_s of inner and outer tubes, showing the metallization trend of DWCNT [13]. Our experimental results for the DWCNT in Figure 2 support the case of *sm*@*sm* in Figure 4a, with a considerable *E*_g_ of ~0.7 eV.

### 2.6. vdW Tunneling Spectroscopy for MWCNTs

Figure 5a shows the transfer curves for various *V*_sd_ at *T* = 4.2 K for an *sm*-MWCNT with *D* ~3 nm and *L* ~2.1 μm (see Appendix A for the AFM and Raman characterizations, respectively); this figure reveals *sm* behavior with an electron (*e*)-doped ambipolar property. In the *e*-doped region (*V*_bg_ > −10 V), the curve shows two current steps at *V*_sd_ = 0.75 V, as indicated by two arrows. With decreasing *V*_sd_, the steps disappear when the saturation current observed for *V*_bg_ > −5 V is lower than those of the current steps. The hole (*h*)-doped region (*V*_bg_ < −20 V) shows rather complicated current modulation behavior. Figure 5b presents a d*I*/d*V*_sd_ map as a function of *V*_sd_ and *V*_bg_. This plot shows several distinct conductance peak curves in both doped regions, as indicated by dashed lines. In the *h*-doped region (*V*_bg_ < −20 V), the *V*_sd_ for a conductance peak curve shows a strong gate dependence, which results in the complicated current modulations observed in this region in Figure 5b. By contrast, the *V*_sd_ for a conductance peak curve in the *e*-doped region is not sensitive to *V*_bg_, except near the depletion region, thereby giving rise to the step-like transfer curve with saturation current observed in the *e*-doped region in Figure 5a. Considering the extension of the energy scale to ~1 eV and the distinctive gate dependence of the conductance peaks, their origin does not lie in the formation of a quantum dot or conductance quantization, as has been suggested previously [14,15]. The white dash-dotted lines in Figure 5b represent the boundaries of the depletion region related to the bandgap of the *sm*-MWCNT. We observed that all of the conductance peak curves bent to lie parallel to the boundaries of the depletion region, which indicates that the conductance peaks could be linked to the DOS related to the electronic band structure of the *sm*-MWCNT, i.e., van Hove singularities in the subband structures. The blue curve in Figure 5c displays d*I*/d*V*_sd_ as a function of *V*_sd_ at *V*_bg_ = 0 V; this plot shows six main conductance peaks, as indicated by inverse triangles. We observed the minimum conductance at *V*_sd_ = 0.1 V, as indicated by the blue arrow in Figure 5c. If we assume that this minimum corresponds to the mid-gap (*E_g_*/2) position, then we can assign two conductance peaks at *V*_sd_ = 35 and 155 mV, which are the peaks nearest to the mid-gap, labeled SC1 and SV1. With this peak assignment, the Fermi level of the CNT (*E*_FC_) is located between SC2 and SC1 at *V*_bg_ = *V*_sd_ = 0 V, as depicted by the vertical dashed line in Figure 5c. The red curve with multiple conductance peaks in Figure 5c was obtained at *V*_bg_ = −40 V, where the minimum conductance was observed at *V*_sd_ ~ 30 mV. The shift in the midgap position in *V*_sd_ with different *V*_bg_ values occurred because the electronic bands were bent by the varying *V*_bg_ values (see Appendix A). When the red curve was shifted to match the minimum conductance to that observed from the blue curve (see Appendix A), the locations of the main conductance peaks in the two curves are nearly coincident, which confirms that the origins of the multiple conductance peaks are the intrinsic electronic states existing in the *sm*-MWCNT, van Hove singularities (also see the band model for sm-MWCNT in Appendix A).

We further examined an *m*-MWCNT with *D* (*L*) ~ 4.8 nm (~1 μm) (see the AFM and Raman characterizations in Appendix A). Figure 6a shows the transfer curve, which exhibits a quasi-metallic behavior. The d*I*/d*V*_sd_ map as a function of *V*_sd_ and *V*_bg_ in the left-hand panel of Figure 6b exhibits multiple pronounced conductance peaks, as indicated by three arrows, which are indicated by the same color-coded arrows, respectively, at *V*_bg_ = 6.2 V in Figure 6c. We noticed a relatively small peak at *V*_sd_ ~ 70 mV for a curve at *V*_bg_ = 6.2 V, as indicated by the blue arrow. After changing the color contrast of the left-hand panel of Figure 6b, we found that the small peak also formed a conductance curve, as indicated by the blue arrow in the right-hand panel of Figure 6b. There were also several weak but finite conductance curves. Our DFT calculations for the metallic DWCNTs in Figure 4c,d show that it is not feasible to assign the peaks even in the calculation results. With this reason, we could not assign the peaks for the *m*-MWCNT in Figure 6. For the *m*-MWCNT case, there should be a finite DOS between the lowest subbands, without a bandgap, which manifests as the finite conductance observed at −0.3 V < *V*_sd_ < 0.3 V in Figure 6c.

## 3. Materials and Methods

### 3.1. Growth of MWCNT

A catalyst liquid composed of CH_3_OH:(Fe(NO_3_)_3_)_9_H_2_O: Al_2_O_3_: MoO2 = 15:20:15:5 (mg) was dropped onto a 500 nm thick SiO_2_/Si substrate with patterned positioning markers and the liquid was blown by nitrogen gas. MWCNTs were grown in a chemical vapor deposition (CVD) chamber with a gas mixture, CH_4_:H_2_ = 5000:500 (sccm), at 915 °C for 10 min.

### 3.2. Theoretical Calculations

We used density functional theory (DFT) methods. All electrons’ Kohn–Sham wave functions were expanded in local atomic orbital basis sets, as implemented in DMOL3 code [16]. The double numeric basis sets with polarization were used and damped atom-pairwise dispersion corrections with the form C_6_R^-6^ using the Tatchenco–Scheffler method were considered [17]. The *k*-points sampled with equidistance of 0.05/Å were used for all calculations to maintain the same criterion among CNTs. We used the generalized gradient approximation (GGA) functional proposed by Perdew–Becke–Ernzerhof (PBE) [18].

### 3.3. Raman and AFM Characterization

The LabRAM HR system (Horiba Scientific) was used for the Raman spectra with a 514 nm wavelength laser and a 100× objective lens (beam exposure time: 10–15 s). The AFM image under tapping mode was obtained with the Park Systems XE-100 AFM.

### 3.4. Experiments

The electrical measurements were performed with a cryo-free system (base temperature: 4 K) in two-probe mode with a voltage source (Keithely 213) and current amplifier (DL 1211). The differential conductance as a function of bias voltage was numerically obtained from the current–voltage characteristics.

## 4. Conclusions

In summary, we performed vdW tunneling spectroscopy for MWCNTs with In metal contacts; *sm*-DWCNT, *sm*- and *m*-MWCNTs, which revealed the electronic DOS of MWCNTs as shown in conventional scanning tunneling spectroscopy [19,20]. The In/MWCNT contact forms a vdW interface, i.e., the electronic DOS of an MWCNT is conserved at the interface, with a vdW vacuum gap. We believe that this is why it is feasible for CNT devices with In metal contacts to provide consistent tunneling spectroscopy. A recent work successfully demonstrated the application of tunneling spectroscopy in SWCNT/hexagonal boron nitride(hBN)/SWCNT heterojunctions, where conductance peaks observed at different *V*_sd_ were interpreted as elastic tunneling into the van Hove singularities of different one-dimensional subbands in both SWCNTs through the hBN tunneling barrier [21]. The transport regime reflecting the van Hove singularities could allow the development of new quantum devices operated by controlled electron (or hole) injection to a specific subband in CNT-based FETs.

## Figures and Tables

**Figure 1 molecules-26-02128-f001:**
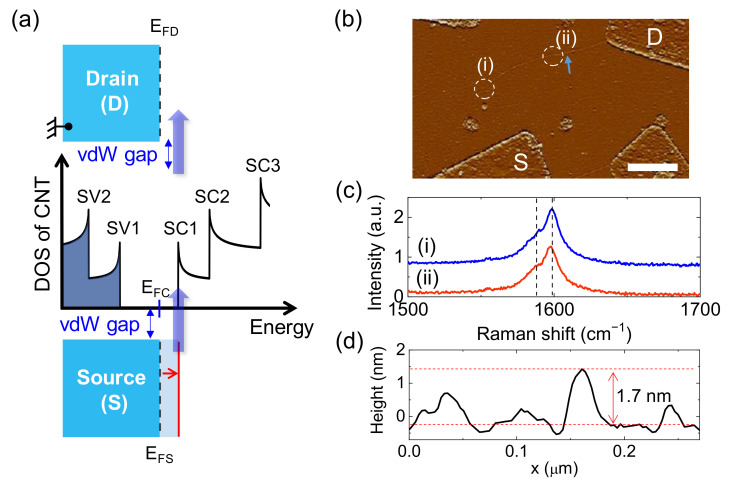
(**a**) Schematic illustration of the electronic DOS of an MWCNT with two indium electrodes. *E*_FD_, *E*_FS_ and *E*_FC_ are the Fermi energies of the drain (D), the source (S) and the CNT, respectively. The solid red lines represent *E*_FS_ at a finite negative *V*_sd_, where *E*_FS_ is aligned with SC1 in the CNT and results in an enhancement of the conductance, as indicated by the blue arrow. (**b**) AFM image of a representative *sm*-DWCNT FET device prepared on a 500 nm thick SiO_2_/Si substrate. The scale bar represents 1 μm. (**c**) Raman signals obtained from circled regions (indicated by (i) and (ii) in (**b**)), which show a main peak at 1598 cm^−1^ as a G peak with a satellite peak at 1586 cm^−1^. (**d**) Height profile along the direction of the blue arrow in (**b**).

**Figure 2 molecules-26-02128-f002:**
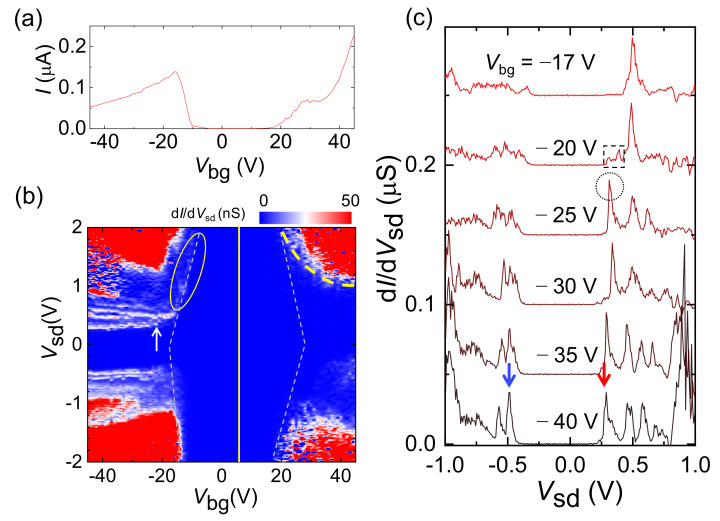
(**a**) *I*-*V*_bg_ (transfer) curve for an *sm*-DWCNT (*L* ≈ 6 μm, *D* ≈ 1.7 nm) at *V*_sd_ = 2 V and *T* = 10 K. (**b**) d*I*/d*V*_sd_ map as a function of *V*_sd_ and *V*_bg_ of the *sm*-DWCT, where the dashed lines indicate the boundaries of the depletion region. The vertical line indicates the midgap location, *E*_g_/2. (**c**) d*I*/d*V*_sd_ as a function of *V*_sd_ for various *V*_bg_ of the *sm*-DWCNT.

**Figure 3 molecules-26-02128-f003:**
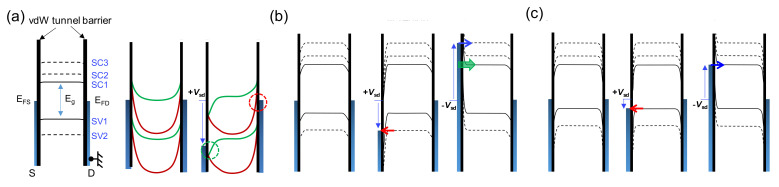
(**a**) Left: electronics bands for *V*_bg_ = 7 V, middle and right: electronic bands for 7 V < *V*_bg_ < 20 V (green curves) and *V*_bg_ > 20 V (red curves). SC*n* (SV*n*) is the *n*th subband in the conduction (valence) band. Electronic bands for (**b**) −20 V < *V*_bg_ < −17 V and (**c**) *V*_bg_ < −22 V.

**Figure 4 molecules-26-02128-f004:**
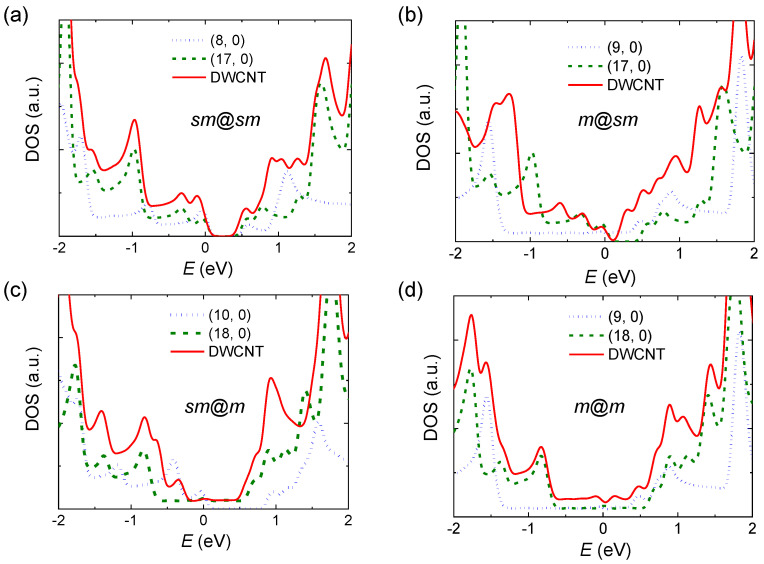
Electronic DOS of DWCNTs, *sm*@*sm* (**a**), *m*@*sm* (**b**), *sm*@*m* (**c**) and *m*@*m* (**d**), which also show the DOS of SWCNTs constructing DWCNTs. Gaussian broadening with 0.05 eV width has been applied to the eigenvalues obtained from DFT calculations.

**Figure 5 molecules-26-02128-f005:**
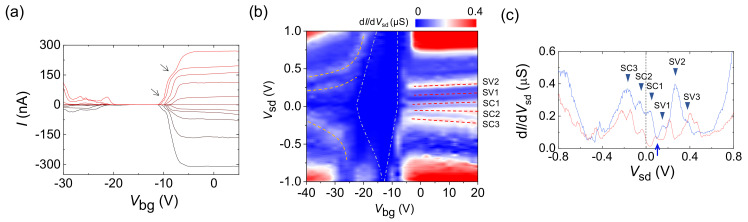
(**a**) Transfer curve for an *sm*-MWCNT (*L* ≈ 2.1 μm, *D* ≈ 3 nm) at *T* = 4.2 K for various *V*_sd_ values from 0.75 V (top) to −0.75 V (bottom) in steps of 0.15 V. Arrows indicate the current steps, originated from the DOS-sensitive transport. (**b**) d*I*/d*V*_sd_ map as a function of *V*_sd_ and *V*_bg_. The dashed curves trace the observed conductance peaks, and the white dash-dotted lines correspond to the boundaries of the depletion region. (**c**) d*I*/d*V*_sd_ as a function of *V*_sd_ at *V*_bg_ = 0 V (blue curve) and −40 V (red curve), where SC*n* (SV*n*) is the *n*th subband in the conduction (valence) band.

**Figure 6 molecules-26-02128-f006:**
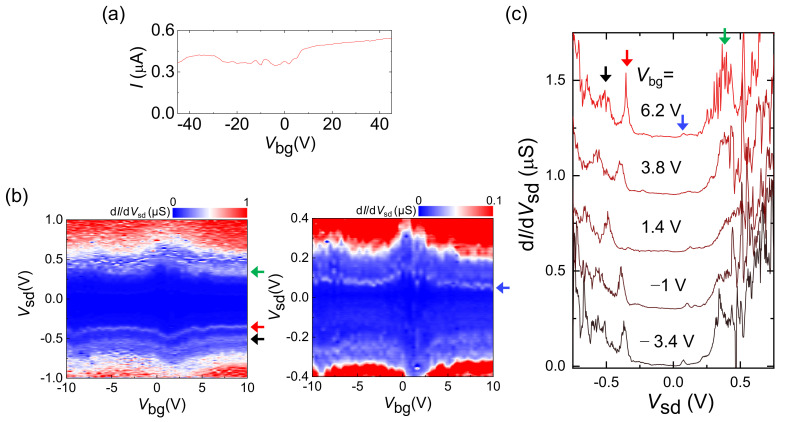
(**a**) Transfer curve for an *m*-MWCNT (*l* ≈ 1 μm, *d* ≈ 4.8 nm) at *V_sd_* = 1 V and *T* = 4.2 K. (**b**) Left panel: d*I*/d*V*_sd_ as a function of *V*_sd_ and *V*_bg_ for the *m*-MWCNT. Right panel: Zoomed plot of the left panel with a different contrast. (**c**) d*I*/d*V*_sd_ as a function of *V*_sd_ for various *V*_bg_ of the *m*-MWCNT.

**Table 1 molecules-26-02128-t001:** Chiralities, diameters, the calculated electronic bandgap (*E*_g_) and the electrical properties of selected SWCNTs.

Chirality	(8, 0)	(9, 0)	(10, 0)	(17, 0)	(18, 0)
Diameter (Å)	6.26	7.05	7.83	13.32	14.10
*E*_g_ (eV)	0.55	0.10	0.91	0.54	0.02
Property	*sm*	(*m*)	*sm*	*sm*	(*m*)

**Table 2 molecules-26-02128-t002:** Chiralities in the inner@outer CNT format, component properties, the calculated electronic *E*_g_ and the electrical properties of DWCNTs composed with SWCNTs listed in Table 1.

Chirality	(8, 0)@(17, 0)	(9, 0)@(17, 0)	(10, 0)@(18, 0)	(9, 0)@(18, 0)
Component Property	*sm*@*sm*	*m*@*sm*	*sm*@*m*	*m*@*m*
*E*_g_ (eV)	0.47	0.22	0.03	0.02
DWCNT property	*sm*	*sm*	(*m*)	(*m*)

## Data Availability

The data presented in this study are available on request from the corresponding author.

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
