# Peer review of "Tunneling Spectroscopy for Electronic Bands in Multi-Walled Carbon Nanotubes with Van Der Waals Gap"

_molecules, 2021, doi:10.3390/molecules26082128_

Round 1

Reviewer 1 Report

In this manuscript, the authors have investigated the possibility to control transport properties of multi-wall carbon nanotubes (MWCNT) using a back-gate bias voltage (Vbg) applied from an Indium (In) electrode. And they measured the transport properties by tunneling spectroscopy methods (I-Vbg curves and dI-dVsd mapping made from many curves measured
at various Vbg ).

I have recognized that main results of this study claimed by the authors are 2 points as described in follows.
1. From results of their tunneling spectroscopy, they successfully obtained results that showed FET-like transport properties of MWCNTs.
2. They also observed dI/dSD curves, in which there were several peaks corresponding to quantum electronic states of CNTs.
Moreover, electron injection to a specific quantum state (sub-band) can be selected by using an appropriate back-gate voltage.

The authors also claimed that MWCNT is one of the useful materials for future application using its unique character (such as semi-conductive and sub-band character). So, this study should be important in the study field of the molecular electronics.

I also think that results of this study include some precious knowledge for realization of future molecular electronic devices.
However, to discuss future application of MWCNT to FET with its quantum electronic states, I think that this manuscript must be revised to be published in “molecules”, because some experimental results and discussions in this manuscript seem to be imperfect. My main questions are 2 as written bellow.

1. Especially, in the 95th line of the manuscript, the authors explained a new dI/dV peak, which appeared around Vsd = 0.3 V with Vbg < -25 V. I wonder how the new peak was induced by such relatively large negative Vbg. The authors wrote that “To understand this, we construct a band model as followed section” in the 100th line of the manuscript. However I could not
find any discussion about this mechanism (on the other hand for negative Vsd, they described from 145th line). I would like the authors to explain “the origin for positive Vsd” in more detail.

2. Relating with quiestion1, in figure5(b) the dI/dV mapping shows 6 sub-bands-states in the Vsd region of -1V < Vsd < +1 V. Among these quantum states, only the “SC3” state moved to opposite way from other states by increasing Vbg. I think this result is interesting and important, because it suggests that the quantum states was influenced (can be controlled) by
application of Vbg. If the authors claim that MWCNT could become a good candidate to be applied for future molecular electronic devices using its quantum transport property, I would like the authors to discuss the reason of the “opposite moving”.

And these follows are just comments.
1. I could not understand the reason why the authors showed results of DFT calculations for several DWCNTs. These CNTs were all zig-zag structure (very special cases).
2. I could not understand the meaning of “Vsd?” in figure5(a).

As written above, I think that this manuscript should be revised for future publication in “molecules” with answer to my questions.
Thank you.

Reviewer 2 Report

This manuscript reports a tunneling spectroscopy study on individual carbon nanotubes with multi-shells. Tunneling contact is of the van der Waals type, which allows the authors to perform the transport spectroscopy over a wide range that is comparable with conventional STM technique. The ability to tune the carrier density by the back-gate gating in conjunction with the bias voltage further provides more spectroscopy information. The paper is very well-written. I also found that the reported data quality is high. Although some of the data interpretation seems to be questionable, I would like to recommend its publication in Molecules if the authors can address my major concerns and some technical details below.

  1. There was a recent paper reporting the tunneling spectroscopy of single-walled carbon nanotubes via a well-defined van der Waals tunneling contact. The authors should cite a recent work, which has reported that [Nano Lett. 2020, 20, 6712−6718]. Applying the same technique in nanotubes with multi-shells do not give rise to new phenomena or physics, but may cause complications due to the contribution from the inner shells and electronic coupling between shells that are likely to occur.
  2. The authors seem to assume only outermost nanotube contributes to the tunneling current. Although they very roughly discussed it in the main text, but can the authors attest to it with their own experimental evidences or from previous reports? Can the authors discuss more about why this is the case?
  3. I am curious how good the Indium-nanotube tunneling contact is in terms of its van der Waals nature that other people [ref. 2] and the authors have used and claimed? The authors said the fermi level was pinned at the contact in their band illustration (Fig. 3), but ref.2 concluded that there was no pinning because of the van der Waals contact formation between Indium and nanomaterials. Can the authors discuss more about it?
  4. The tunneling scheme the authors used in the experiment actually forms a double tunneling barrier in source and drain contacts. This was not clearly mentioned in the main text, in particular, in the schematic of Fig. 1a.
  5. The assignment of peaks: One noticeable example is that the authors denied the SV1 and SV2 tunneling events and assigned the first main peak at positive side to the SV3 (Figure 2 data), but they allow them to happen when describing data of figure 5 & figure 6. This (including others) poses question mark to me about the data interpretation.
  6. Calculation in Table 1&2: First, I do not think these calculations can prove the inner nanotube gives negligible contribution to the tunneling current, which is also related to my second question. Second, why the calculated band gap of (9,0)@(17,0) becomes larger than that of the isolated (9,0)? We know that moiré effect can play a role, which seems not considered in their calculation. That’s fine. Can the authors provide more insights on the origin of the band gap enlargement in their calculation?
  7. Gate efficiency: Judging from the charge neutrality shift in Fig. 5c, it suggests a very low gate efficiency. This is often seen in practice. I am wondering what the authors think about the causing in their experiments.

Round 2

Reviewer 1 Report

My reviewer's comments are attached.
